# Medication Reconciliation of Patients by Pharmacist at the Time of Admission and Discharge from Adult Nephrology Wards

**DOI:** 10.3390/pharmacy12060170

**Published:** 2024-11-18

**Authors:** Hossein Ahmadi, Yalda Houshmand, Ghanbar Ali Raees-Jalali, Iman Karimzadeh

**Affiliations:** 1Student Research Committee, School of Pharmacy, Shiraz University of Medical Sciences, Shiraz P.O. Box 7146864685, Iran; hoseinahmadi@sums.ac.ir; 2Department of Clinical Pharmacy, School of Pharmacy, Shiraz University of Medical Sciences, Shiraz P.O. Box 7146864685, Iran; yaldahushmand@sums.ac.ir; 3Department of Internal Medicine, School of Medicine, Shiraz University of Medical Sciences, Shiraz P.O. Box 7134814336, Iran; raeesjgh@sums.ac.ir

**Keywords:** drug errors, drug interactions, medication discrepancies, medication reconciliation, nephrology

## Abstract

Purpose: The aim of the present study was to investigate the impact of medication reconciliation by pharmacists at both admission and discharge in hospitalized patients with different kidney diseases. Methods: A prospective study was performed in adult nephrology wards of a teaching referral hospital in Iran from September 2020 to March 2021. All patients hospitalized in the nephrology ward for at least 1 day who received the minimum of one medication during their ward stay within the study period were considered eligible. Medication reconciliation was performed by taking a best-possible medication history from eligible patients during the first 24 h of ward admission. Medications were evaluated for possible intentional as well as unintentional discrepancies. Results: Here, 178 patients at admission and 134 patients at discharge were included. The mean numbers of unintentional drug discrepancies for each patient at admission and discharge were 6.13 ± 4.13 and 1.63 ± 1.94, respectively. The mean ± SD numbers of prescribed medications for patients before ward admission detected by the nurse/physician and pharmacist were 6.06 ± 3.53 and 9.22 ± 4.71, respectively (*p* = 0.0001). The number of unintentional discrepancies at admission and discharge had a significant correlation with the number of drugs used and underlying diseases. The number of unintentional discrepancies at admission was also correlated with patients’ age. The number of comorbidities was significantly associated with the number of unintentional medication discrepancies at both admission and discharge. At the time of ward discharge, all patients were given medication consultations. Conclusions: The rate of reconciliation errors was high in the adult nephrology ward. The active contribution of pharmacists in the process of medication reconciliation can be significantly effective in identifying these errors.

## 1. Introduction

Medication errors (MEs) are one of the main causes of morbidity and mortality in both outpatient and inpatient settings [1]. ME is defined as “any preventable event that may lead to inappropriate medication use or patient harm while the medication is in the control of the health care professional, patient, or consumer” [2]. It is estimated that approximately 1.5 million people in the United States are potentially exposed to harm from ME each year [3]. The annual cost of MEs in the European zone has been estimated to be EUR 4.5–21.8 billion [4].

MEs can occur at various stages of pharmacotherapy, including prescribing, transcribing/documenting, dispensing, administering, and monitoring [5]. The UK’s National Patient Safety Agency has reported that MEs occur at a rate of almost 50% in administration, 18% in dispensing, and 16% in prescribing stages [6]. A systematic review of MEs in Iran indicates that the prevalence levels of prescribing, transcribing, dispensing, and administering MEs are 47.8%, 51.8%, 33.6%, and 70%, respectively [7]. Patient care transfer is one of the most common causes of MEs, including changes in setting, service, practitioner, or level of care. During hospitalization, changing the patients’ physician within a certain ward or following ward transfer may lead to alterations in their prescription, including the addition of new drugs, the omission of current ones, or the changing of medication dose/frequency of administration [8]. In this regard, about half (40 to 60%) of the MEs in hospitals are estimated to occur on admission into or discharge from a clinical unit or hospital, or even at the transfer of care, and around 30% of these errors have the potential to cause the patient harm [9]. In patients with chronic diseases or several co-morbidities such as end-stage kidney disease (ESKD), this problem becomes more critical, because they often take at least ten to twelve medications to slow the progression of the disease and also treat associated complications in different organs/systems, such as cardiovascular, dermatologic, hematologic, endocrine, and neurologic [10].

Medication discrepancies are defined as incompatibilities between two or more drug lists. In other words, any differences between the patient’s previous medication regimen and the list of drugs administered at admission, discharge, or transmission are considered medication discrepancies. The severity and type of these discrepancies vary, and include skipping medications regularly used, prescribing another drug of the same category, having differences in drug dosage forms or routes of administration, and changing the dose or time interval [11,12]. Medication discrepancies are either unintentional or intentional. Generally, unintentional discrepancies need intervention through written and/or verbal communications [13]. Medication discrepancies can lead to an increased hospital stay, the need for a post-discharge emergency home visit, re-hospitalization, and the usage of additional healthcare services [14]. Overall, medication discrepancies affect around 70% of patients at admission, transfer, or discharge; one-third of which have the potential to harm patients [15,16].

Medication reconciliation is the process of identifying a precise list of drugs taken by a patient and comparing it to his/her current medications. It can help reduce different MEs, including omission, doubling, dosing, and interaction, leading to improved pharmacotherapy and patient safety [17]. Therefore, performing appropriate medication reconciliation can decrease MEs and consequent adverse effects by up to 75% [18]. Comprehensive and appropriate medication reconciliation can be achieved by taking a best-possible medication history (BPMH) [19,20]. Patients at a higher risk of reconciliation errors, such as the elderly who receive several medications (polypharmacy) and those having long-term hospitalization, are likely to gain more benefits from the medication reconciliation process [21].

Considering the undeniable importance of performing medication reconciliation under different healthcare conditions, and considering the lack of adequate published data in Iranian inpatient settings, an interventional study was designed to assess different aspects of medication reconciliation performed by pharmacists at both admission and discharge stages of patients on the nephrology wards of a referral teaching hospital in Iran.

## 2. Material and Method

### 2.1. Study Design, Setting, Duration

The present investigation was prospective and interventional. It was performed in the adult nephrology wards of Namazi Hospital, which features two independent wards (Ward 1 and Ward 2) with a total of 36 beds and a bed occupancy rate of more than 90%. The study lasted for 6 months, from September 2020 to March 2021. The ethical committee of the Shiraz University of Medical Sciences, Shiraz, Iran, approved this study (ethical ID: IR.SUMS.REC.1399686). The study process was briefly and simply described for each patient or his/her family members. They were assured that identifying/confidential patient data would not be shared. Finally, a written, signed/dated, informed consent was taken from patients or their family members.

### 2.2. Patients Selection and Medication History Obtaining

Eligibility criteria for this study were all patients hospitalized in the nephrology ward for at least 1 day who received the minimum of one medication during their ward stay within the study period. Therefore, no certain sample size was considered and calculated. In addition, there were no limitations regarding age, sex, type of disease, comorbidities, or class/number of medications for recruiting patients (convenience sampling).

On all weekdays, except for holidays and weekends, a trained pharmacist under the supervision of a clinical pharmacist performed medication reconciliation at both the admission (during the first 24 h of ward admission) and discharge stages (immediately before ward discharge). The retroactive model of medication reconciliation was used in this study. Accordingly, physicians wrote admission orders for medications in the cohort before the BPMH was taken by the pharmacist. This medication ordering was mostly based on patients’ admission diagnosis, underlying disease(s), and medical history taken by physicians/nurses at the time of ward admission.

The medication reconciliation form used in this study includes the following contents: Demographic and anthropometric characteristics (age, sex, weight, height, body mass index [BMI], initial diagnosis, comorbidities, duration of ward stay, and medications [received before admission and those prescribed by the physician during the ward stay]). The glomerular filtration rate (GFR) of patients with no acute kidney injury (AKI) at ward admission was calculated by the Chronic Kidney Disease Epidemiology Collaboration (CKD-EPI) 2009 serum creatinine-based formula: 141 × min(Scr/κ, 1)α × max(Scr/κ, 1) − 1.209 × 0.993Age [×1.018 if female] [×1.159 if black], where Scr is serum creatinine, κ is 0.7 for females and 0.9 for males, α is −0.329 for females and −0.411 for males, min is the minimum of Scr/κ or 1, and max is the maximum of Scr/κ or 1 [22]. In patients with chronic kidney disease (CKD), their disease was categorized into 1 of 5 stages based on GFR values [23].

The patient’s medications list was completed based on the BPMH taken by the pharmacist. This lists encompass all the patient’s medications including prescription drugs, over-the-counter medications, herbal drugs, and medicinal plants. Accordingly, apart from face-to-face interviews between the pharmacist and the patient, at least one other reliable source of information was used to obtain and verify all prescribed and non-prescribed medications such as herbal/natural products and vitamins/minerals/supplements taken by the study population before hospital admission. The complementary medicine information sources used in this study included family/relatives patients’ referral letters to hospitals, any previous hospital discharge summary, patients’ own medication lists, and patients’ copies of their repeated prescriptions. These sources were used by the pharmacist to take BPMH from patients during the first 24 h of ward admission.

### 2.3. Medication Discrepancies Assessment and Reporting

As a part of the medication reconciliation process performed by the pharmacist, the patient’s medical history before admission was compared with prescribed medications at the time of ward admission and discharge. Any medication discrepancies were divided into two categories: intentional and unintentional. Intentional medication discrepancies were those that had a certain cause or justification from a physician’s perspective. Unintentional discrepancies had no rationale and they were divided into six ME categories, as follows: 1—omission (deletion of a prescribed medication), 2—commission (administration of a new medication that is not needed), 3—dose, frequency or route of administration, 4—documentation (mistake in recording the prescribed medication, the dosage, or the frequency), 5—doubling (prescribing two or more medications from the same class) and 6—substitution (stopping a medication without a reason and substituting it by another drug from the same class). Unintentional medication discrepancies identified by the pharmacist upon admission and discharge were classified based on the Anatomical Therapeutic Chemical/Defined Daily Dose (ATC/DDD) Index 2021 [24]. They were recorded in the medication reconciliation form and copies of completed forms with the pharmacist’s signature were added into the patients’ medical charts. In addition, the in-charge physicians were informed about identified unintentional medication discrepancies of their patients by the pharmacist in-person at the earliest time.

### 2.4. Potential Drug–Drug Interaction Evaluations

In addition to reconciliation errors, potential moderate to major (type D or X) drug–drug interactions (DDIs) between medications at the time of discharge were also assessed by the pharmacist using the Lexi-Interact online software (Lexi-Interact Online 2021) [25]. The mechanism and type of DDIs were also determined.

### 2.5. Patient Consultation at the Time of Ward Discharge

At the time of ward discharge, all patients were provided medication consultations (dietary considerations, appropriate dose, route, and time interval of administration along with required monitoring, common adverse effects, potential DDIs, and storage condition) by the pharmacist. This medication consultation was offered to patients both verbally and non-verbally.

In summary, interventions implemented by the pharmacist in this study involved taking BPMH along with identifying intentional/unintentional discrepancies as a part of medication reconciliation, detecting potential type D or X DDIs, and consulting patients about their medications at the time of ward discharge.

### 2.6. Study Endpoints

The total number of medication discrepancies was considered as the primary endpoint. The number of prescribed medications for patients before ward admission recorded by the pharmacist versus physicians/nurses, the number of either intentional or unintentional medication discrepancies, the types of unintentional medication discrepancies, and the number and type of potential DDIs were determined as secondary endpoints.

### 2.7. Statistical Analyses

The SPSS software (version 20, SPSS Institutes, Chicago, IL, USA) was used for all data analyses. Categorical data were described as number and percent. The normal distribution of continuous variables was assessed by the Kolmogorov–Smirnov test. Normally and non-normally distributed continuous variables were defined as mean ± standard deviation (SD) and median (interquartile range), respectively. The Mann–Whitney test was performed to compare the number of medications given before ward admission that were identified by the pharmacist with those detected by nurses/physicians. The Spearman test was used to determine the possible correlation of the number of unintentional medication discrepancies at ward admission as well as discharge with continuous variables (age, BMI, GFR, duration of ward stay, and number of co-morbidities). A linear regression model was exploited to assess the possible association between the numbers as well as the classes of unintentional medication discrepancies and different demographic/clinical features (age, BMI, GFR, duration of ward stay, number of co-morbidities, and number of medications). The mean or median numbers of medication classes at ward admission and discharge were compared by the Wilcoxon test. This test was also used to compare the mean or median number of unintentional medication discrepancies at ward admission and ward discharge. Finally, the Kruska–Wallis test was performed to evaluate the possible association between different stages of CKD (only in patients with CKD) and the mean number of unintentional medication discrepancies at both ward admission and discharge. In all the above tests, *p*-values less than 0.05 were considered statistically significant.

## 3. Results

### 3.1. Patients’ Characteristics

Medication reconciliations for 178 and 134 patients were performed at ward admission and discharge, respectively. The following demographic and clinical data are related to 178 subjects. More than half of the cohort (104 out of 178, 58.4%) were male. According to Table 1, the mean age of the study population fit into the middle age category. Based on the median (interquartile range) of GFR calculated by the CKD-EPI formula at ward admission, CKD patients mostly belonged to stage 5. The length of hospital stay of the participants ranged from 2 to 53 days. The most common initial diagnosis was AKI (34.27%), followed by ESKD complications, such as catheter-related bloodstream infection (15.17%) and nephrotic syndrome (12.92%). Hypertension (71.3%), CKD (55.1%), and diabetes mellitus (36.5%) were the most frequent underlying diseases in the participants.

### 3.2. Medication Discrepancies and Medication Reconciliation

Based on interviews performed by nurses/physicians and the pharmacist, the mean ± SD numbers of prescribed medications for patients before ward admission were 6.06 ± 3.53 and 9.22 ± 4.71, respectively. This difference was statistically significant (*p*-value = 0.0001). In addition, the mean ± SD number of prescribed medications to the study cohort at the time of ward discharge was 7.81 ± 3.06.

The total numbers of unintentional as well as intentional medication discrepancies, along with the different types of unintentional medication discrepancies, are listed in Table 2. At ward admission, 167 out of 178 patients (93.82%) had at least one unintentional medication discrepancy. This rate at the time of ward discharge was 84 out of 134 patients (62.68%). The median (IQR) total numbers of medication discrepancies per patient were 9.58 (5.61) cases at ward admission and 5.01 (3.13) cases at ward discharge (*p*-value < 0.0001). Similarly, the median number of unintentional medication discrepancies was significantly higher at ward admission compared to that at ward discharge (5.91 cases and 1 case, respectively [*p*-value < 0.0001]).

The total numbers of medication discrepancies (*p*-value < 0.0001), unintentional errors (*p*-value < 0.0001), omission errors (*p*-value < 0.0001), doubling errors (*p*-value = 0.013), documentation errors (*p*-value < 0.0001), and dosing errors (*p*-value = 0.002) showed a significant difference between admission and discharge times. In contrast, there was no significant difference in the median of discrepancies related to commission errors and substitution errors between admission and discharge (*p*-value > 0.05).

Regarding drug classes related to unintentional medication discrepancies at ward admission (Table 3), categories A (24.26%, 1.94 ± 1.56), C (19.5%, 1.75 ± 1.12), and V (17.41%, 1.17 ± 0.98) were the most common. Calcium salts, nitrocantine, and gabapentin were the drugs most frequently associated with omission errors at ward admission. In terms of doubling errors at ward admission, the co-administrations of prednisolone with either methylprednisolone or hydrocortisone and Nephrovit^®^ (a multivitamin for patients with CKD) with folic acid were the most frequent. The most common documentation errors at ward admission were related to herbal medicines (e.g., senna, thyme), hyoscine, and dicyclomine. At ward admission, dosing errors, commission errors, and substitution errors were mostly associated with atorvastatin, pantoprazole, and pantoprazole (instead of famotidine), respectively. At the time of ward discharge, category A (26.17%, 0.79 ± 0.40) was the most frequent drug class related to unintentional medication discrepancies, followed by category C (17.8%, 0.62 ± 0.28) and category B (15.7%, 0.49 ± 0.23). At the time of ward discharge, the most common omission, doubling, documentation, dosing, commission, and substitution errors belonged to calcium salts and gabapentin, Nephrovit^®^ with folic acid, brimonidine and latanoprost, atorvastatin and nitrocantine, pantoprazole and hyoscine, and losartan (instead of valsartan), respectively. Except for drug classes L and P, the mean ± SD numbers of unintentional medication discrepancies in other drug categories were statistically significant at the time of ward admission and ward discharge (*p*-value < 0.05).

According to Table 4, the total number of unintentional medication discrepancies significantly correlated with patient age (r = 0.263, *p*-value < 0.0001), number of comorbidities (r = 0.417, *p*-value < 0.0001), and the number of administered medications (r = 0.358, *p*-value < 0.0001) at the time of ward admission. At the time of ward discharge, the total number of unintentional medication discrepancies significantly correlated with a number of comorbidities (r = 0.325, *p*-value < 0.0001) and the number of administered medications (r = 0.339, *p*-value < 0.0001).

Based on linear regression analysis (Table 5), the number of unintentional medication discrepancies at ward admission has a significant association with the number of comorbidities (*p*-value = 0.043) and the number of administered medications (*p*-value = 0.023). At the time of ward discharge, only the number of comorbidities was significantly associated with the number of unintentional medication discrepancies (*p*-value = 0.031) (Table 5). According to the Kruskal–Wallis test, the numbers of unintentional medication discrepancies at ward admission (*p*-value = 0.166) as well as discharge (*p*-value = 0.660) were comparable among different stages of CKD.

### 3.3. Potential Drug–Drug Interactions

At the time of ward admission, type-D or -X potential DDIs were detected in 97 out of 178 (59.49%) patients. In addition, 79 out of 134 (58%) patients experienced at least one type-D or -X potential DDI at the time of ward discharge. The mean ± SD total numbers of potential DDIs per patient on ward admission and discharge were 1.17 ± 0.93 and 0.91 ± 0.59, respectively. The characteristics of the 10 most commonly identified type-D and all type-X potential DDIs in the cohort are listed in Table 6. The most frequent type-D potential DDI in the cohort was heparin + acetylsalicylic acid (30 cases), followed by prednisolone + calcium carbonate (17 cases) and mycophenolate + calcium carbonate (12 cases). Atorvastatin + cyclosporine (3 cases) was the most prevalent type-X potential DDI in the study population.

### 3.4. Medication Consultation at Ward Discharge and Acceptance Rate of Pharmacist Interventions

Medication counseling and required drug information were offered to all 134 patients at the time of ward discharge. These included 83 medications. All patient consultations were provided both verbally and non-verbally (on written forms). The acceptance rate of the pharmacist interventions for medication reconciliation at ward admission and ward discharge by nurses/physicians was about 82%.

## 4. Discussion

### 4.1. General Concepts and Findings

In this study, medication reconciliation at both admission and discharge was conducted by the pharmacist in the adult nephrology inpatient ward. Herein, the nephrology setting was selected because there are inadequate data published in this field, and patients with kidney disease (particularly CKD) are at risk of reconciliation errors, mostly due to their disease state complexities (multiple comorbidities) and also complicated medication regimens. The number of patients that received medication reconciliation at ward admission (178 cases) was higher than that at ward discharge (134 cases) in the present investigation. This difference is attributed to possible death in the ward or the transfer/discharge of patients on holidays/weekends when the pharmacist was not available.

### 4.2. Medication Discrepancies and Medication Reconciliation

In our study, the pharmacist interviews identified significantly more pre-admission drugs used by patients than the nurse/physician interview did. In accordance with our survey, similar findings were also reported in a prospective and observational study in the geriatric ward of a hospital in Belgium [26]. Regarding unintended medication discrepancies at the time of hospital admission, Cornish et al. showed that pharmacists can identify and report all prescription and non-prescription drugs (100%), which compares favorably to the levels detected by the physician (79% and 45% of all prescription and non-prescription drugs, respectively) [27]. The incomplete medical histories obtained by physicians/nurses may be due to the heavy workload of these members of health-care team, especially during the COVID-19 pandemic peaks, and also having inadequate knowledge about the different pharmacokinetics and pharmacodynamics properties of medications [28].

A large number of unintentional medication discrepancies was identified at the times of both ward admission (93.82%) and ward discharge (62.68%). In addition, documentation and omission errors at ward admission and omission errors at ward discharge were the most common reconciliation errors in this cohort. In another study in the same hospital, at seven wards, including neurology, internal, and emergency, Karimzadeh et al. revealed that the major cause (85.8%) of unintentional medication discrepancies at the time of ward admission was omission errors [29]. Similarly, data from four other studies at teaching hospitals in Iran also show that the most common types of MEs were omission ones [30,31,32,33]. In line with these findings, Prins et al. also reported that 78% of patients admitted to a psychiatric clinic in the Netherlands had at least one discrepancy, of which 69% were omission errors [34]. According to a prospective, observational study undertaken at the hemodialysis unit of a tertiary care university-affiliated teaching hospital in Toronto, 338 out of 512 (66%) medication discrepancies were unintentional [35]. In contrast, the rate of medication discrepancy at admission to the internal medicine, cardiology, and general surgery departments of a teaching hospital in the United States was only 23% [36]. More recently, a retrospective, single-center study in a chronic hemodialysis unit in the United States revealed that medication reconciliation by at least two clinical pharmacists identified unintentional discrepancies, undocumented intentional discrepancies, and medication-related problems in 53%, 71%, and 59% of encounters, respectively [37]. Finally, the results of a systematic review of 42 quantitative and qualitative studies involving 3,755 patients show that medication discrepancies occurred in 10% to 67% of patients, and 60% to 67% of subjects reported at least one omission or commission error [38]. The disparity in the rate of medication discrepancies reported in different studies can be attributed to a variety of factors, as follows: the definition of medication discrepancy, the number of patients admitted and discharged from the hospital (patient turnover), the knowledge level and attitude of physicians/nurses about the importance and role of medication reconciliation, and the level of pharmacist participation in performing medication reconciliation.

Although reconciliation errors can occur at any stage of patient care, our current research focused on the time points of hospital admission and hospital discharge. A number of studies have found that 95% of patients at the time of admission to the hospital and 73% of individuals at hospital discharge may experience MEs [10,18]. Interestingly, this study demonstrated that the number of unintentional medication discrepancies at the time of ward discharge (62.68%) was much lower than that at ward admission (93.82%). This difference can be partially explained by the fact that a clinical pharmacist and his residents regularly and actively attended the educational and therapeutic rounds of the nephrology ward, and reviewed the pharmacotherapy of each patient and providing corrective feedback. Similarly, a retrospective study conducted in a nephrology ward of the National Taiwan University Hospital demonstrated that the number of medication reconciliations significantly increased (from 4 to 119) after the participation of clinical pharmacists [39].

The drug categories associated with the most common medication discrepancies in our study population were alimentary tract and metabolic systems, blood and blood-forming organs, and cardiovascular drugs. A prospective, cross-sectional study in the nephrology ward of an academic referral hospital in Iran demonstrated that most of the MEs were associated with cardiovascular drugs [31]. Chan et al. reported that commissions and incorrect dose/frequency errors in an academic hemodialysis unit in Toronto, Canada, mainly related to blood pressure medications or phosphate binders [35]. A retrospective, observational cohort study on nephrology patients at the Leiden University Medical Center, the Netherlands, revealed that the alimentary tract and metabolism (20.4%), the cardiovascular system (15.4%), and the nervous system (12.2%) were most commonly involved with medication transfer errors. Medication discrepancies with antihypertensives, antibiotics, antidiabetic drugs, opioid analgesics, antipsychotics, anticoagulants, and immunosuppressive classes can cause more damage to patients [40].

An increasing number of underlying diseases usually have a direct connection with aging, and often lead to an increase in the number of drugs taken by the patient. According to the results of the Spearman and linear regression analyses, the number of unintentional medication discrepancies either at the time of ward admission or discharge had a significant correlation and association with patient age, the number of underlying diseases, and the number of given medications in our cohort. In particular, this was the case for patients with CKD and ESKD under hemodialysis, who were prescribed an average of 6 to 8 and 12 medications, respectively [41]. Geurts et al., in a retrospective study in the Netherlands, showed that the number of medication discrepancies at the time of ward discharge had a significant association with the number of patient medications at this time point [42]. In line with this, the results from a pilot and prospective study in the internal medicine and surgery wards of a teaching hospital in Saudi Arabia suggested that medication discrepancies had a significant association with the patient’s age (over 65 years) and polypharmacy [43]. Similarly, a prospective study in the endocrine–diabetic and nutrition departments of a hospital in France indicated that the number of drugs taken by patients, identified by the clinical pharmacist, had a significant association with medication discrepancies at admission and discharge [44]. In addition, a retrospective cohort study on medication reconciliation at admission and discharge in a tertiary care academic teaching hospital in the United States reported age as one of the risk factors for medication discrepancies on admission [36]. Finally, Ebbens et al. identified that the number of medications per patient was significantly associated with medication transfer errors at an outpatient transplant nephrology clinic in the Netherlands (OR = 1.11, 95%CI = 1.04–1.16; *p*-value < 0.05) [40].

### 4.3. Potential Drug-Drug Interactions

In our study, heparin + acetylsalicylic acid was the most common type-D potential DDI both at ward admission and discharge. The concurrent use of these two medications can increase the risk of bleeding. This potential DDI can mostly be managed by dose adjustments of both medications and regularly monitoring the signs and symptoms of bleeding. However, this combination has been recommended as a viable option for the management of acute coronary syndromes, and the real clinical relevancy of this DDI is questionable at the bedside. This is also true for mycophenolate + cyclosporine potential DDI. This interaction can potentially lead to reduced bioavailability, which consequently diminishes the efficacy of mycophenolic acid, secondary to blocking the enterohepatic cycle of mycophenolic acid by cyclosporine [45]. Physicians should be aware about this DDI, and reduce the daily dose of mycophenolic acid by about 500 mg when switching from cyclosporine to tacrolimus in order to minimize mycophenolic acid’s dose-dependent adverse effects, such as gastrointestinal disorders and myelosuppression [46,47]. Notably, the mycophenolate + cyclosporine combination is commonly used in certain kidney diseases, such as glomerulonephritis, or as a maintenance immunosuppression regimen of kidney transplantation.

Atorvastatin + cyclosporine was the most frequent type-X potential DDI in the present investigation. Similarly, Moradi et al. reported atorvastatin + cyclosporine as the most common type-X DDI in a cross-sectional study in outpatient kidney transplant patients in Iran [48]. Its mechanism is described as increasing the serum concentration of atorvastatin by cyclosporine via inhibiting CYP3A4-dependent enzymes, as well as reducing the hepatic uptake of atorvastatin by OATP1B1-SLCO1B1. This could result in myopathy, rhabdomyolysis, and finally AKI. Since therapeutic drug monitoring for atorvastatin is not currently available in many centers worldwide, limiting the dose of atorvastatin to no more than 10 mg/day in patients receiving cyclosporine concomitantly appears to be a more realistic and practical approach to prevent or minimize complications of this DDI [48].

### 4.4. Acceptance Rate of the Pharmacist Interventions

Our study revealed that the acceptance rate of all the pharmacist interventions as medication reconciliation at both admission and discharge by the healthcare team was about 82%. The major reason behind the non-acceptance of pharmacist intervention in the present investigation was that some physicians were not directly in charge of patients, and so they did not take the full responsibility of their medication prescriptions at ward admission or discharge. This phenomenon is expected in teaching hospitals, where different types of physicians, including interns, residents, fellows, and attending, may be simultaneously involved with the diagnostic and therapeutic processes of a certain patient. The other cause of rejecting pharmacist interventions about medication reconciliation was that some physicians in the nephrology ward were either too cautious or unwilling to apply required changes in the medication regimen of patients that had been previously prescribed in the emergency unit, or even before hospital admission.

Interestingly, according to the results of a prospective interventional study in a university-affiliated clinic in Iran, 131 out of 145 (90.34%) recommendations suggested by the clinical pharmacist regarding identified unintentional discrepancies were both accepted and applied by the clinicians [32]. These rates are within the range (from 41.91% to 94.5%) reported in a systematic review of 39 articles in clinical pharmacists’ interventions in Iran published in December 2018 [49]. The acceptance rates of clinical pharmacists’ interventions reported from European and American studies are 73–89% and 85–99%, respectively [50]. The possible challenges with implementing interventions, such as medication reconciliation by pharmacists to improve medication adherence in the setting of CKD and dialysis, have been discussed in detail elsewhere [51,52].

### 4.5. Study Limitations

One of the major limitations of our study is that the severity and clinical importance of the detected reconciliation errors were not determined. The lack of a control group in our investigation precluded the assessing of the clinical and economic significance of the implemented medication reconciliations. The possibilities of internal validity problems due to center and selection biases could not be ruled out in this investigation. Although different information sources were used to take BPMH, this process should not be considered flawless because there was no access to some reliable sources, such as the community pharmacy dispensing history of recruited patients. The other likely pitfall is that a retroactive type of BPMH registration was used in the present survey. A proactive model, which relies on physicians taking BPMH prior to writing admission medication orders, may be more effective, practical, and timely. In this regard, a prospective, quasi-experimental study at the cardiology unit of an academic hospital in Iran found that the numbers of patients with unintentional discrepancies as well as unintentional medication discrepancies were significantly lower in proactive compared to retroactive cases. Interestingly, the duration of the medication reconciliation process was also significantly shorter in the proactive compared to the retroactive model [53]. Finally, our patients were not followed after ward discharge to evaluate the possible long-term, positive impacts of our medication reconciliation and patient counseling.

## 5. Conclusions

The current study shows that more than 90% and 60% of patients at the time of nephrology ward admission and discharge experienced at least one unintentional medication discrepancy, respectively. The average number of medication discrepancies at ward admission was significantly higher compared to that at the time of ward discharge. The numbers of patient comorbidities and co-administered medications were significantly linked to unintentional medication discrepancies at both ward admission and discharge. Pharmacists identified a significantly higher number of medications prescribed to patients before admission compared to those detected during medical history taking by nurses or physicians. Medication counseling was offered to all 134 patients at the time of ward discharge.

## 6. Clinical Implications and Future Perspectives

The implementation of the medication reconciliation program as a regular practice, preferably by pharmacists, in different phases of patient care in the hospital is crucial. The importance of the pharmacist’s role in this process is highlighted by the fact that pharmacists have the highest level of knowledge and experience related to different aspects of medications (pharmacology, pharmacotherapy, pharmacokinetics, toxicology, pharmaceutical technology, pharmaceutical chemistry, pharmacognosy) among healthcare professionals.

Concerning inadequate trained pharmacists, along with this being a quite time-consuming process and the presence of a high patient turn-over rate, especially in referral teaching hospitals, individuals at higher risk of reconciliation errors such as those with advanced age, numerous comorbidities (e.g., CKD), under polypharmacy, and receiving high-alert medications (e.g., antithrombotic) should be given priority in performing medication reconciliation. This can assist pharmacists in stratifying and prioritizing patients that need and may benefit more from the medication reconciliation process. Determining the long-term clinical and economic impacts of medication reconciliation for patients with different kidney diseases after hospital discharge can be a research topic for future investigations in this area.

## Figures and Tables

**Table 1 pharmacy-12-00170-t001:** Demographic/clinical characteristics and primary diagnoses of the study population at ward admission.

Variables	Median	Interquartile Range	Minimum	Maximum
Age (year)	58.00	29.00	18	97
Body mass index (kg/m^2^)	22.92	5.42	2.00	68.00
Length of hospital stay (Day)	9.00	7.00	2.00	53.00
Glomerular filtration rate for non-AKI patients	14.005	25.10	2.90	97.28
Primary diagnosis	Number		Percent	
Acute kidney injury	61		34.27	
End-stage kidney disease complications	27		15.17	
Nephrotic syndrome	23		12.92	
Urinary tract infection/sepsis/catheter site infection	21		11.8	
Kidney transplant rejection	12		6.74	
Post-transplant complications other than transplant rejection	12		6.74	
Other	9		5.06	
Lupus nephritis	7		3.93	
Diabetic nephropathy in chronic kidney disease (stage 1 to 4)	5		2.81	

**Table 2 pharmacy-12-00170-t002:** The numbers and types of different medication discrepancies during ward admission and discharge in the study population.

Types of Unintentional Medication Discrepancies	Ward Admission		Ward Discharge		*p*-Value
	Median	Interquartile Range	Median	Interquartile Range	
Total number of medication discrepancies	9.58	5.61	5.01	3.13	<0.0001
Number of unintentional medication discrepancies	5.91	3.79	1	3	<0.0001
Omission error	1.75	1.63	1	2	<0.0001
Doubling error	0.12	0.33	0.04	0.19	0.013
Documentation error	3.86	2.59	0.07	0.43	<0.0001
Dosing error	0.22	0.47	0.11	0.34	0.002
Commission error	0.05	0.24	0.04	0.23	0.99
Substitution error	0.13	0.35	0.09	0.31	0.06

**Table 3 pharmacy-12-00170-t003:** The distribution of the total number of medication discrepancies per patient at ward admission and discharge in relation with the WHO drug classification system.

Drug Category		Ward Admission		Ward Discharge		*p*-Value
		Mean ± Standard Deviation	Number (Percent)	Mean ± Standard Deviation	Number (Percent)	
A	Alimentary Tract and Metabolism	1.94 ±1.56	255 (24.26)	0.79 ±0.40	50 (62.17)	<0.0001
B	Blood And Blood Forming Organs	1.09 ± 0.61	110 (10.46)	0.49 ± 0.23	30 (15.7)	<0.0001
C	Cardiovascular System	1.75 ± 1.12	205 (19.5)	0.62 ± 0.28	33 (17.8)	<0.0001
D	Dermatological	0.29 ± 0.04	8 (0.66)	0.00 ± 0.00	0	<0.0001
G	Genito-Urinary System and Sex Hormones	0.41 ± 0.11	20 (1.9)	0.12 ± 0.01	2 (1.4)	0.006
H	Systemic Hormonal Preparations, Excl. Sex Hormones and Insulins	0.51 ± 0.22	35 (3.33)	0.21 ± 0.04	5 (2.61)	<0.0001
J	Anti-infectives For Systemic Use	0.52 ± 0.16	28 (2.66)	0.28 ± 0.05	6 (3.14)	0.006
L	Antineoplastic And Immunomodulating Agents	0.29 ± 0.08	14 (1.33)	0.19 ± 0.04	4 (2.9)	0.13
M	Musculo-Skeletal System	0.47 ± 0.20	38 (2.61)	0.12 ± 0.01	2 (1.4)	<0.0001
N	Nervous System	1.10 ± 0.52	100 (9.51)	0.59 ± 0.21	27 (41.13)	<0.0001
P	Antiparasitic Products, Insecticides and Repellents	0.00 ± 0.00	0 (0)	0.00 ± 0.00	0 (0)	NA
R	Respiratory System	0.79 ± 0.17	28 (2.66)	0.17 ± 0.01	2 (1.4)	0.07
S	Sensory Organs	0.57 ± 0.17	28 (2.66)	0.39 ± 0.04	5 (2.61)	0.002
V	Various	1.17 ± 0.98	183 (17.41)	0.41 ± 0.19	24 (21.56)	<0.0001

**Table 4 pharmacy-12-00170-t004:** The correlation between the number of drug discrepancies and demographic/clinical characteristics at ward admission and discharge.

Variable	Number of Medication Discrepancies at Ward Admission		Number of Medication Discrepancies at Ward Discharge	
	Coefficient Correlation (r)	*p*-Value	Coefficient Correlation (r)	*p*-Value
Age	0.263	<0.0001	0.084	0.350
Length of hospital stay	0.076	0.312	0.102	0.241
Body mass index	0.138	0.088	0.083	0.357
Glomerular filtration rate	0.152	0.051	−0.033	0.708
Number of underlying diseases	0.417	<0.0001	0.325	<0.0001
Number of administered medications	0.358	<0.0001	0.339	<0.0001

**Table 5 pharmacy-12-00170-t005:** Linear regression analysis of the possible associations between the number of medication discrepancies at ward admission and discharge and the demographic/clinical characteristics of the study population.

Variable	Number of Medication Discrepancies at Admission		Number of Medication Discrepancies at Discharge	
	Regression Coefficient (β)	*p*-Value	Regression Coefficient (β)	*p*-Value
Age	0.018	0.348	−0.003	0.788
Gender	0.364	0.577	0.014	0.970
Length of hospital stay	0.018	0.676	0.004	0.892
Body mass index	0.051	0.313	0.022	0.422
Glomerular filtration rate	−0.012	0.762	0.005	0.815
Number of underlying diseases	0.478	0.043	0.229	0.031
Number of administered medications	0.230	0.023	0.072	0.072

**Table 6 pharmacy-12-00170-t006:** The most common type-D and all type-X potential drug–drug interactions at the time of ward admission and discharge in the study population.

DDI	Severity of DDI	Number of Patients (Percent)	Number of DDIs at Admission (Percent)	Number of DDIs at Discharge (Percent)
Type D				
Heparin + Acetylsalicylic acid	Moderate	30 (33.7)	30 (33.7)	0
Prednisolone + Calcium carbonate	Moderate	17 (19.10)	14 (15.73)	16 (20.25)
Mycophenolate + Calcium carbonate	Moderate	12 (13.48)	10 (11.23)	11 (13.92)
Ciprofloxacin + Calcium carbonate	Moderate	7 (7.86)	4 (4.49)	3 (3.80)
Mycophenolate + Cyclosporine	Moderate	7 (7.86)	6 (6.74)	6 (7.59)
Acetylsalicylic acid + Apixaban	Major	5 (5.62)	1 (1.12)	5 (6.33)
Levothyroxine + Calcium carbonate	Moderate	5 (5.62)	5 (5.62)	4 (5.06)
Heparin + Piracetam	Moderate	4 (4.49)	4 (4.49)	0
Vancomycin + Piperacillin-tazobactam	Major	4 (4.49)	4 (4.49)	0
Atorvastatin + Diltiazem	Major	3 (3.37)	3 (3.37)	0
Type X				
Atorvastatin + Cyclosporine	Major	3 (3.37)	1 (1.12)	3 (3.80)
Calcitriol + Vitamin D	Major	2 (2.25)	2 (2.25)	0
Apixaban + Heparin	Major	1 (1.12)	1 (1.12)	0
Apixaban + Carbamazepine	Major	1 (1.12)	1 (1.12)	0
Spironolactone + Triamterene/hydrochlorothiazide	Major	1 (1.12)	0	1 (1.26)
Potassium chloride + Fexofenadine	Moderate	1 (1.12)	1 (1.12)	0
Potassium chloride + Quetiapine	Moderate	1 (1.12)	1 (1.12)	0
Tiotropium bromide + Ipratropium bromide	Major	1 (1.12)	1 (1.12)	0
Salbutamol + Carvedilol	Major	1 (1.12)	1 (1.12)	0
Salmeterol-fluticasone + Carvedilol	Major	1 (1.12)	1 (1.12)	0

## Data Availability

The data that support the findings of this study are available from the corresponding author upon reasonable request. Identifying/confidential patient data will not be shared.

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
