# Peer review of "Medication Reconciliation of Patients by Pharmacist at the Time of Admission and Discharge from Adult Nephrology Wards"

_pharmacy, 2024, doi:10.3390/pharmacy12060170_

Round 1
Reviewer 1 Report
Comments and Suggestions for Authors
The manuscript titled Medication Reconciliation of Patients by Pharmacist at 2 the Time of Admission and Discharge from Adult 3 Nephrology Wards by Ahmadi et al. is well written and interesting. The authors depict clearly the importance of the presence of clinical pharmacists on nephrology wards.
I have some minor issues I would like the authors to clearify - There is a discrepancy of the number of patients reviewed on admission compared to discharge. I think it would be clearer if this were to be explained in the text. Moreover, you use sometimes average sometimes median values of parameters, I understand the importance of knowing the normality of distribution but it is not usual to see both approachs in one table. I suggest you opt for the fact that the distribution was not normal for all parameters and you use median and interquartile values in tables.
Comments on the Quality of English Language
English is good, there are some minor mistakes in the text.
Author Response
â– Comments from reviewer 1:
Comment 1: The manuscript titled Medication Reconciliation of Patients by Pharmacist at 2 the Time of Admission and Discharge from Adult 3 Nephrology Wards by Ahmadi et al. is well written and interesting. The authors depict clearly the importance of the presence of clinical pharmacists on nephrology wards.
Response 1: Thanks for your complement.
Comment 2: There is a discrepancy of the number of patients reviewed on admission compared to discharge. I think it would be clearer if this were to be explained in the text.
Response 2: Thank you for pointing this out. The possible reasons behind the difference between number of patients received medication reconciliation at ward admission (n=178) and that at ward discharge (n=134) were provided in the first paragraph of the discussion section (lines 317 & 318).
Comment 3: You use sometimes average sometimes median values of parameters, I understand the importance of knowing the normality of distribution but it is not usual to see both approaches in one table. I suggest you opt for the fact that the distribution was not normal for all parameters and you use median and interquartile values in tables.
Response 3: Thank you for pointing this out. To avoid any misunderstanding and confusion, all continuous variables in tables 1 & 2 of the result section (pages 5 & 6) were just expressed as median and interquartile range.
Comment 4: English is good, there are some minor mistakes in the text.
Response 4: Thank you for pointing this out. The manuscript was reviewed again and all grammatical and spelling errors have been corrected.

Reviewer 2 Report
Comments and Suggestions for Authors
1) The Authors performed an interesting article entitled "Medication reconciliation of patients by pharmacist at the time of admission and discharge from adult nephrology wards". The aim of the study was to investigate the impact of medication reconciliation by pharmacists at both admission and discharge in hospitalized patients with different kidney diseases. Overall, the publication has clinical interest, is well organized and structured, was approved by an Ethics Committee, and contains statistical analysis of the results.
2) Please do not use the trade name "Aspirin®" instead of the international non-proprietary name of the active substance "acetylsalicylic acid". The commercial name has no scientific meaning.
3) Lines 217-218: Regarding the "Calcium supplements", I am not sure whether they were medicines or dietary supplements. Please clarify this aspect.
4) Please avoid using "we" in the article. For instance, replace “We aimed to investigate the impact of medication reconciliation…" by “The aim of this article was to investigate the impact of medication reconciliation…".
5) Lines 113-120: Place the equation properly (separate from the text).
6) A criterion should be applied to bibliographic references, that is, either put the "DOI" in all of them or not put it in any.
7) The pharmacist is the health professional with the most knowledge about medicines (that is, about Pharmacology, Pharmacotherapy, Pharmacokinetics, Toxicology, Pharmaceutical Technology, Pharmaceutical Chemistry, Pharmacognosy, etc.), allowing them to be used rationally and safely. In this way, the patient would only benefit if this professional had a more active role in terms of pharmacotherapy/medication reconciliation. Therefore, I think that this aspect should be further highlighted in the article.
Author Response
â– Comments from reviewer 2:
Comment 1: The Authors performed an interesting article entitled "Medication reconciliation of patients by pharmacist at the time of admission and discharge from adult nephrology wards". The aim of the study was to investigate the impact of medication reconciliation by pharmacists at both admission and discharge in hospitalized patients with different kidney diseases. Overall, the publication has clinical interest, is well organized and structured, was approved by an Ethics Committee, and contains statistical analysis of the results.
Response 1: Thanks for your complement.
Comment 2: Please do not use the trade name "Aspirin®" instead of the international non-proprietary name of the active substance "acetylsalicylic acid". The commercial name has no scientific meaning.
Response 2: Thanks for pointing this out. The term “aspirin” was replaced by "acetylsalicylic acid" throughout the manuscript (lines 291 & 406) and table 6 (pages 9 & 10).
Comment 3: Lines 217-218: Regarding the "Calcium supplements", I am not sure whether they were medicines or dietary supplements. Please clarify this aspect.
Response 3: Thanks for pointing this out. To avoid any misunderstanding, the term "calcium supplements" have been replaced by “calcium salts” throughout the manuscript.
Comment 4: Please avoid using "we" in the article. For instance, replace “We aimed to investigate the impact of medication reconciliation…" by “The aim of this article was to investigate the impact of medication reconciliation…".
Response 4: Thanks for pointing this out. All sentences within the manuscript with the term “we” were rephrased to remove it.
Comment 5: Lines 113-120: Place the equation properly (separate from the text).
Response 5: Thanks for pointing this out. The CKD-EPI 2009 creatinine formula in the method section (lines 122 to 125) was separated from the text.
Comment 6: A criterion should be applied to bibliographic references, that is, either put the "DOI" in all of them or not put it in any.
Response 6: Thanks for pointing this out. If available, we added DOI for all references. If it is not available, the PMID of the article was added in the reference list.
Comment 7: The pharmacist is the health professional with the most knowledge about medicines (that is, about Pharmacology, Pharmacotherapy, Pharmacokinetics, Toxicology, Pharmaceutical Technology, Pharmaceutical Chemistry, Pharmacognosy, etc.), allowing them to be used rationally and safely. In this way, the patient would only benefit if this professional had a more active role in terms of pharmacotherapy/medication reconciliation. Therefore, I think that this aspect should be further highlighted in the article.
Response 7: Thanks for pointing this out. We are agreed with you and added an individual section after conclusion entitled “Clinical implications & Future perspectives” to address several issues like the importance of pharmacist role in the process of medication reconciliation (lines 486 to 491).

Reviewer 3 Report
Comments and Suggestions for Authors
Interesting study in Iran especially where no other information is available. The major concern is with the study methodology.
Materials and Methods need significant improvement. What is your primary outcome variable as well as any secondary variables.? What was your hypothesis? Need to define your variables such as intentional versus unintentional discrepancies.
Was not clear how the study design was done. When did pharmacists intervene versus physician or nurses. Hard to follow the methods and interventions, not clearly written.
You have a convenience sample here and that needs to be described.
In your study design, someone only staying 24 hours is not an accurate reflection of pre and post intervention.
Not sure the rationale for running a regression analysis--there are too many confounding issues. Older patients will likely have more medications and more disease states. What were you trying to prove with this regression? What was our outcome variables (dependent variable) versus independent variables? Why linear regression versus logistic regression?
You should be reporting medians with IQR rather than mean and SD. Medication errors are categorical and not continuous data. You can not have 1/2 of a medication error. These are all whole numbers and ordinal in nature. Table 3 for example
So much information but it left you confused at times. Simplify and describe for someone not involved in the study.
Watch English in the paper. A number of errors.
Many of your drug interactions would be considered acceptable therapy for some patients. Example calcium and thyroid--they can be given in the same day but just separated. Thyroid would be given in the AM and calcium later in the day. Disagree with many of your drug-drug interactions.
Explain why not all pharmacist recommendations were accepted. Reasons?
There are many more limitations than stated by authors. Significant internal validity issues.
Conclusion should be limited to what you found in the study specifically.
Comments on the Quality of English Language
Some significant English issues that need corrected.
Author Response
â– Comments from reviewer 3:
Comment 1: Interesting study in Iran especially where no other information is available. The major concern is with the study methodology.
Response 1: Thanks for your complement.
Comment 2: Materials and Methods need significant improvement. What is your primary outcome variable as well as any secondary variables.? What was your hypothesis? Need to define your variables such as intentional versus unintentional discrepancies.
Response 2: Thank you for pointing this out. An individual subsection was added into the method section to describe primary and secondary endpoints (lines 172 to 178). Intentional and unintentional discrepancies have been defined in the method section (lines 143 to 145).
Comment 3: Was not clear how the study design was done. When did pharmacists intervene versus physician or nurses. Hard to follow the methods and interventions, not clearly written.
Response 3: Thanks for pointing this out. The “study type” was provided in the method section (line 92). All interventions implemented by pharmacists in the study were described individually in the method section (lines 168 to 171).
Comment 4: You have a convenience sample here and that needs to be described.
Response 4: Thanks for pointing this out. This term was added into the method section (line 106).
Comment 5: In your study design, someone only staying 24 hours is not an accurate reflection of pre and post intervention.
Response 5: Thanks for pointing this out. The “minimum-maximum duration of hospital stay” was added into table 1 of the result section. It ranges from 2 to 53 days (lines 207 & 208). Notably, staying at least 24 hour in the ward is one of the inclusion criteria that has been mentioned in the method section. We are agreed with you that if a patient stayed in the ward for a longer period, the impact of medication reconciliation at ward admission versus hospital discharge appear to be more sensible and prominent.
Comment 6: A criterion should be applied to bibliographic references, that is, either put the "DOI" in all of them or not put it in any.
Response 6: Thanks for pointing this out. If available, we added DOI for all references. If it is not available, the PMID of the article was added in the reference list.
Comment 7: Not sure the rationale for running a regression analysis--there are too many confounding issues. Older patients will likely have more medications and more disease states. What were you trying to prove with this regression? What was our outcome variables (dependent variable) versus independent variables? Why linear regression versus logistic regression?
Response 7: Thanks for pointing this out.
We are agreed that the presence of several independent variables may negatively affect the prediction of the model, especially if the sample size will be large enough. As you know, regression analysis is a standard statistical approach in clinical studies for evaluating possible associations. We used this test to assess the possible association of different variables as an independent (demographic, clinical, laboratory) with a dependent variable (the total number of medication discrepancies) to have a better and more realistic view about the possible independent risk factors of medication discrepancies. As we mentioned in the “Clinical Implications & Future perspectives” section (lines 485 to 501), due to the lack of required trained pharmacist along with being quite time-consuming and high patient turn-over rate especially in referral, teaching hospitals, individuals at higher risk of reconciliation errors such as those with advanced age, numerous comorbidities (e.g., CKD), under polypharmacy, and those receiving high alert medications (e.g., anti-thrombotics) should give priority in performing medication reconciliation. Therefore, identifying risk factors can assist the pharmacists in stratifying and prioritizing patients that may need and benefit more from the medication reconciliation process.
Regarding the type of statistical test, considering the fact that the independent variable (the total number of medication discrepancies) is a continuous quantitative variable, it seems that the appropriate statistical test is “linear regression”, rather than “logistic regression”. As you know, the latter is suitable for testing associations where there is one dichotomous dependent variable (in two categories).
Comment 8: You should be reporting medians with IQR rather than mean and SD. Medication errors are categorical and not continuous data. You can not have 1/2 of a medication error. These are all whole numbers and ordinal in nature. Table 3 for example
Response 8: Thanks for pointing this out. To avoid any misunderstanding and confusion, all continuous quantitative variables in tables 1 & 2 of the result section were expressed just as “median and interquartile range” (pages 5 & 6). Related sentences in the result section were also changed (lines 228 to 231).
Comment 9: So much information but it left you confused at times. Simplify and describe for someone not involved in the study.
Response 9: Thanks for pointing this out. We are agreed with you that the volume of data and findings in this study are considerable. We tried to classify and provide them with an order that can be followed easily and useful for the reader. In addition, we summarized the major and more practical findings of our study in the conclusion section.
Comment 10: Watch English in the paper. A number of errors.
Response 10: Thank you for pointing this out. The manuscript was reviewed again and all grammatical and spelling errors have been corrected.
Comment 11: Many of your drug interactions would be considered acceptable therapy for some patients. Example calcium and thyroid--they can be given in the same day but just separated. Thyroid would be given in the AM and calcium later in the day. Disagree with many of your drug-drug interactions.
Response 11: Thank you for pointing this out. We are agreed with you that a number of identified drug-drug interactions in our study have not enough clinical relevancy and importance. Therefore, a “potential” term was added before the drug-drug interaction throughout the manuscript to avoid any misunderstanding. In addition, a paragraph was also added in the discussion section (lines 409 to 420) about mycophenolate + cyclosporine potential drug-drug interaction to address and highlight this issue.
Comment 12: Explain why not all pharmacist recommendations were accepted. Reasons?
Response 12: Thank you for pointing this out. The major reasons of non-acceptance of pharmacist interventions by physicians in the context of medication reconciliation were provided and explained in the discussion section (lines 434 to 443).
Comment 13: There are many more limitations than stated by authors. Significant internal validity issues.
Response 13: Thank you for pointing this out. A sentence was added in “Study Limitations” subsection to address internal validity issue (lines 458 & 459).
Comment 14: Conclusion should be limited to what you found in the study specifically.
Response 14: Thank you for pointing this out. The conclusion section was reduced. We also added a new section after conclusion entitled “Clinical implications & Future perspectives” to address practical issues and research topics of medication reconciliation in patients with kidney diseases (lines 485 to 501).

Reviewer 4 Report
Comments and Suggestions for Authors
This is a work of great interest, with great application and consequences for hospital care, but rarely addressed and described adequately. Although the work is carried out on a small population, and with characteristics not entirely extrapolated to other populations, it also has a quality and wealth of data that make it a very attractive manuscript.
The title is brief, but descriptive and appropriate to the theme and design of the study.
The abstract is of adequate length, presents subdivision into sections, shows the most important contents of each segment of the manuscript, and presents the main numerical results with clear and concise data.
The introduction is a bit long, but describes very well both the concepts, definitions and definitions used as well as the prevailing conceptual reality on the subject, providing a theoretical basis that supports the work and clearly expresses the objectives of the text.
The materials and methods are described in great detail, considering all the methodological, process, statistical, and analysis methodology aspects used. The level of detail allows the reproducibility of the work and ensures its correct evaluation.
The results are expressed with appropriate numerical data, with appropriate measures of association and statistical significance. Correct statistical tests are used for the variables analyzed, and tables that are too long are used.
It is suggested that the long row titles be reduced and that some format be used that allows the longitudinal length of the tables to be reduced in order to improve their appreciation. The content of the tables is appropriate.
The discussion is of appropriate length, does not present any significant biases, and shows specific data from other studies that allow the comparison of the results obtained. The causes of the findings are analyzed, and the limitations and applications of the results obtained are described.
The conclusion is too long (it seems more like an extension of the discussions), and shows specific data without significant biases, but it should be reduced.
Finally, I must say that although I do not find any important methodological or content objections, the manuscript suffers from a lack of projection of the real usefulness of the work. It is not clear in any definitive way (although the authors try to describe it in the discussion) how evaluating these discrepancies really is useful and how this process should be continuously monitored. In short, it is clear that the authors have a definite guiding idea that they did not fully capture in the text. A brief paragraph should be included in the discussion on "future directions" suggesting the implementation of this type of process as a routine work mechanism and showing its usefulness in the "continuous improvement" of health care.
In summary, the work does not present a need for major revisions (in my opinion), but it requires minor corrections that would significantly improve the final message.
Author Response
â– Comments from reviewer 4:
Comment 1: This is a work of great interest, with great application and consequences for hospital care, but rarely addressed and described adequately. Although the work is carried out on a small population, and with characteristics not entirely extrapolated to other populations, it also has a quality and wealth of data that make it a very attractive manuscript.
Response 1: Thanks for your complement.
Comment 2: The title is brief, but descriptive and appropriate to the theme and design of the study.
Response 2: Thanks for your complement.
Comment 3: The abstract is of adequate length, presents subdivision into sections, shows the most important contents of each segment of the manuscript, and presents the main numerical results with clear and concise data.
Response 3: Thanks for your complement.
Comment 4: The introduction is a bit long, but describes very well both the concepts, definitions and definitions used as well as the prevailing conceptual reality on the subject, providing a theoretical basis that supports the work and clearly expresses the objectives of the text.
Response 4: Thanks for your complement.
Comment 5: The materials and methods are described in great detail, considering all the methodological, process, statistical, and analysis methodology aspects used. The level of detail allows the reproducibility of the work and ensures its correct evaluation.
Response 5: Thanks for your complement.
Comment 6: The results are expressed with appropriate numerical data, with appropriate measures of association and statistical significance. Correct statistical tests are used for the variables analyzed, and tables that are too long are used.
Response 6: Thanks for your complement.
Comment 7: It is suggested that the long row titles be reduced and that some format be used that allows the longitudinal length of the tables to be reduced in order to improve their appreciation. The content of the tables is appropriate.
Response 7: Thanks for your complement. Some subtitles in tables are revised and shortened.
Comment 8: The discussion is of appropriate length, does not present any significant biases, and shows specific data from other studies that allow the comparison of the results obtained. The causes of the findings are analyzed, and the limitations and applications of the results obtained are described.
Response 8: Thanks for your complement.
Comment 9: The conclusion is too long (it seems more like an extension of the discussions), and shows specific data without significant biases, but it should be reduced.
Response 9: Thanks for your complement. The conclusion section was reduced (lines 473 to 483).
Comment 10: Finally, I must say that although I do not find any important methodological or content objections, the manuscript suffers from a lack of projection of the real usefulness of the work. It is not clear in any definitive way (although the authors try to describe it in the discussion) how evaluating these discrepancies really is useful and how this process should be continuously monitored. In short, it is clear that the authors have a definite guiding idea that they did not fully capture in the text. A brief paragraph should be included in the discussion on "future directions" suggesting the implementation of this type of process as a routine work mechanism and showing its usefulness in the "continuous improvement" of health care.
Response 10: Thanks for pointing this out. We add a new section after conclusion entitled “Clinical implications & Future perspectives” to address practical issues and research topics of medication reconciliation in patients with kidney diseases (lines 485 to 501).

Round 2
Reviewer 3 Report
Comments and Suggestions for Authors
Thank you for thoughtful consideration of reviewer comments.
Comments on the Quality of English Language
consider deleting "on the other hand" from the document.
Author Response
Response: Please find the attached file that"on the other hand" term has been either removed or replaced throughout the manuscript based on the reviewer 3 comment.